# Fermented Diet Liquid Feeding Improves Growth Performance and Intestinal Function of Pigs

**DOI:** 10.3390/ani11051452

**Published:** 2021-05-19

**Authors:** Huailu Xin, Mingyu Wang, Zou Xia, Bing Yu, Jun He, Jie Yu, Xiangbing Mao, Zhiqing Huang, Yuheng Luo, Junqiu Luo, Hui Yan, Huifen Wang, Quyuan Wang, Ping Zheng, Daiwen Chen

**Affiliations:** 1Institute of Animal Nutrition, Sichuan Agricultural University, Chengdu 611130, China; xinhuailu@stu.sicau.edu.cn (H.X.); wmy970602@163.com (M.W.); xiazou1001@163.com (Z.X.); ybingtian@163.com (B.Y.); hejun8067@163.com (J.H.); jerryyujie@163.com (J.Y.); acatmxb2003@163.com (X.M.); z.q.huang@163.com (Z.H.); luoluo212@126.com (Y.L.); 13910@sicau.com (J.L.); yan.hui@sicau.edu.cn (H.Y.); wanghuifen1005@163.com (H.W.); ice_5885327@163.com (Q.W.); 2Key Laboratory of Animal Disease-Resistant Nutrition, Chengdu 611130, China

**Keywords:** fermented liquid feeding, growth performance, intestinal function, pigs

## Abstract

**Simple Summary:**

The present study indicated that fermented liquid feeding improved the growth performance of pigs, which might be associated with gastrointestinal hormone and intestinal functions. These results provided a new perspective for improving the growth performance of pigs.

**Abstract:**

Accumulating evidences demonstrate that fermented feed and liquid feeding exerted a great beneficial influence on growth performance and health in the pig industry. This experiment was conducted to evaluate the effects of fermented liquid feeding on the growth performance and intestinal function of pigs. Two hundred and eighty-eight 27-day-old weaned piglets (8.21 ± 0.27 kg) were randomly allocated to a control group (basal diet (CON)), an antibiotic group (basal diet supplemented with antibiotics (AB)) and a fermented liquid feeding group (basal diet with fermented liquid feeding (FLF)), with 6 replicates per treatment and 16 weaned piglets per replicate. The experiment lasted for 160 days. Fresh fecal samples were collected to evaluate the apparent total tract digestibility (ATTD) of nutrients from the last 4 days of each stage. The results are shown as follows: (1) Compared with the CON group, in the whole stage, the FLF diet significantly increased the final body weight (BW) and ADG of pigs (*P* < 0.05), and had a tendency to increase ADFI (*P* = 0.086), but had no effect on F/G. (2) The ATTD of dry matter (DM), crude protein (CP), ether extract (EE), crude ash (CA), crude fiber (CF), gross energy (GE), calcium (Ca) and total phosphorus (TP) in the FLF group was significantly elevated compared with those of the CON group at 8–20 kg stage (*P* < 0.05). Meanwhile, the ATTD of EE in the FLF group was significantly increased compared with that of the CON group at the 50–75 kg and 100–125 kg stages (*P* < 0.05), and the ATTD of Ca was higher than that of CON group at the 100–125 kg stage (*P* < 0.05). (3) Compared with that of the CON group, the level of serum leptin in the FLF group had a tendency to decrease (*P* = 0.054), the level of serum ghrelin in the FLF group was significantly elevated (*P* < 0.05) and the level of serum peptide YY in the FLF group was significantly decreased (*P* < 0.05). (4) The abundance of *Lactobacillus* in cecal and colonic digesta was observably enhanced in FLF group. Meanwhile, the abundance of *Escherichia coli* in cecal and colonic digesta were dramatically reduced in the FLF group compared with that in the CON and AB groups (*P* < 0.05). (5) The levels of acetic acid in colonic digesta were significantly increased in the FLF group (*P* < 0.05), and an increasing trend was observed in total VFA in colonic digesta compared with CON (*P* < 0.1). The levels of acetic acid in colonic digesta were significantly promoted in the FLF group compared with that of the AB group (*P* < 0.05). In conclusion, these results indicate that fermented liquid feeding improved the growth performance of pigs, which might be associated with gastrointestinal hormone and intestinal functions.

## 1. Introduction

In 2006, the European Union completely banned the application of antibiotic additives in pig diets. The Ministry of Agriculture and Rural Affairs of China issued the same regulation for pigs in 2020. It is hard to completely replace antibiotics with just an improved formula; thus, it is imperative to seek more healthy and efficient measures to exert synergistic effects on formula.

Liquid feeding, a widely used technique that mixes water with feed at a constant ratio [1], has been proven to exert a beneficial influence on the growth performance of pigs, including the generation of increases in ADG and ADFI [2,3,4]. However, high labor intensity and the waste of liquid feeding mixed with hand are not conducive to its popularization, which could be solved by special-made liquid feeding equipment. Special-made liquid feeding equipment has the advantages of providing a higher degree in automation and lower feed waste compared with liquid feeding mixed by hand.

The interest in fermented feeds for improving the growth performance of pigs increased dramatically after the announcement of the ban on the use of antibiotics as antimicrobial growth promoters for swine in the European Union. Fermented feeds been extensively studied due to their benefits of increasing nutrient bioavailability and nutritional value [5]. Fermented feeds have been shown to have beneficial effects on the growth performance of pigs [6,7,8]. Previous studies mostly focused on the fermentation of single raw materials, such as single grain or soybean meal [9,10,11]. It is believed that the use of a complete diet for fermentation can further improve the nutritional value of feed. However, there is no relevant report on the fermentation of compound raw materials.

Limited experimental results demonstrate the superiority of fermented liquid feeding in weaned piglets [12,13,14]. However, fermented liquid feeding has only been reported in weaned piglets, and no research on fermented liquid feeding in finishing pigs has been reported. Therefore, this experiment was conducted to study the effects of fermented liquid feeding on the growth performance, nutrient digestibility, serum hormones, intestinal digesta flora and microbial metabolites of pigs.

## 2. Materials and Methods

### 2.1. Preparation of Fermented Feed and Its Quality Determination

The enzyme preparation used in this experiment was obtained from Guangdong VTR Bio-Tech Co., Ltd., and the strains were purchased from China General Microbiological Culture Collection Center (CGMCC).

The composition of fermented feed was consistent with the basal feed at each stage. The fermentation process was developed according to patent (no.201910736307.0).

The basel diet and fermented diet were dried at 65 °C for 72 h, after which they were ground to pass through a 40-mesh screen. All feed samples were measured for dry matter, crude protein and crude fat [15], crude fiber [16], Ca [17] and P [18]. Gross energy was determined using a specific adiabatic oxygen bomb calorimetry (Parr Instrument Co., Moline, IL, USA).

Approximately 1 g feed samples were used to determine pH. Briefly, the supernatants of feed samples were centrifuged at 6000× *g* for 5 min after adding 10 mL distilled water (PHS-3C, INESA, Shanghai, China).

Lactic acid in feed were analyzed using commercial kits (Nanjing Jiancheng Bioengineering Institute, Nanjing, China) combined with a UV-VIS Spectrophotometer (UV1100, MAPADA, Shanghai, China) according to the manufacturer’s instructions. Acid-soluble proteins were determined [19].

### 2.2. Experimental Design and Animal Management

Experimental procedure and animal care were accomplished in accordance with the guide for the care and use of laboratory animals provided by the Institutional Animal Care Advisory Committee for Sichuan Agricultural University. A total of two hundred and eighty-eight 27-day-old pigs (Duroc × Landrace × Yorkshire, weaned at 27 d) with body weights of 8.21 ± 0.27 kg were used in a 160-d experiment, and all pigs remained on the same dietary treatments throughout the trial. At the beginning of the experiment, weaned pigs were randomly assigned to three treatments with six replicates on the basis of their initial BW and sex. The three treatments were CON (control, dry fed basal diet), AB (antibiotic, dry fed basal diet +50 mg/kg Kitasamycin, 75 mg/kg chlortetracycline) and FLF (fermented liquid feeding, the basal diet was fed after fermentation). What follows is additional explanation for Kitasamycin: It is a multicomponent macrolide antibiotic produced by Streptomyces. It has an inhibitory effect on Gram-positive bacteria and Gram-positive bacteria, especially for most penicillin and red mold Vegetarian Staphylococcus aureus, for which it is effective.

The basal diet was formulated to meet or exceed the nutrient requirements recommended by the NRC (2012). The ingredients and compositions of the basal diet are presented in Table 1. The experiment was divided into the nursery period and the fattening period. In the nursery period (8–20 kg), each pen (4.0 × 3.0 m) came equipped with a slatted plastic floor and four stainless-steel nipple drinkers. In the growing–finishing period (20–125 kg), each pen (8.0 × 3.0 m) was also equipped with a slatted plastic floor and four stainless-steel nipple drinkers. All pigs were fed diets four times per day at 0800, 1200, 1600 and 2000 for a 160-d period. Water was provided ad libitum to pigs. We fermented the feed the day before we used it. The FLF diet was prepared by mixing fermented diet with water at a ratio of 2.5 L water per kg feed by an automatic liquid feeding device (HHIS-010, Henan He shun Automation Equipment Co., Ltd., Deng feng, China). Troughs were emptied daily and residual feed was weighed back. Any fouled feed was removed, dried and weighed to estimate wastage. The wasted feed was also collected, dried and weighed to estimate wastage. All pigs were weighed at the beginning and the end of each phase after 12 h of fasting, and feed intake per pen was recorded daily throughout the experiment to calculate ADG, ADFI and F/G.

### 2.3. Sample Collection

Samples of the dry basal diet and fermented feed were collected for chemical analysis. Fresh fecal samples were collected from 16 pigs per pen immediately after defecation and then placed in individual plastic bags. After each collection of feces, 10 mL of a 10% H_2_SO_4_ solution was added to each 100 g of wet fecal sample for the fixation of excreta nitrogen. All feed and fecal samples were stored at −20 °C until analysis.

On day 161, prior to the morning feeding and following overnight fasting, one pig with the average BW in each pen was chosen and bled. Blood samples were collected from the precaval vein into nonheparinized vacuum tubes and then centrifugated (3500× *g* for 10 min at 4 °C), following which serum samples were collected and stored at −20 °C for serum parameters analysis. The abdomen was opened in the laminar airflow clean benches, and the small intestine, cecum and colon were removed immediately. The samples from the middle sections (4 cm) of the duodenum, jejunum and ileum were collected and stored in 4% fresh paraformaldehyde solution for histomorphology measurements. The tissues of the duodenum, jejunum and ileum were gently chopped with a knife and snap-frozen in liquid nitrogen and then stored at −80 °C for further analyses. In addition, the digesta from the middle cecum (10 cm) and middle colon (10 cm) were collected and stored at −80 °C to measure the microbial quantity and microbial metabolites.

### 2.4. Measurement of ATTD of Nutrients

Feces from the last 4 d of each stage of each pen were mixed thoroughly and dried at 65 °C for 72 h, after which they were ground to pass through a 40-mesh screen. ATTD was measured using acid insoluble ash (AIA) as the internal marker. The content of AIA in feed and fecal were measured according to the Chinese National Standard [20]. The samples were analyzed for DM (method 930.15; AOAC), ash (method 923.03; AOAC), EE (method 920.39; AOAC) and CP (method 990.03; AOAC), according to the Association of Official Analytical Chemists [15], and CF (GB/T, 6434, 2006), Ca (GB/T, 6436, 2018) and P (GB/T, 6437, 2018) were analyzed according to the Chinese National Standard. The gross energy of feed and fecal was determined using a specific adiabatic oxygen bomb calorimetry (Parr Instrument Co., Moline, IL, USA). The ATTD was calculated using the following formula: ATTD (%) = {1 − [(A1 × F2)/(A2 × F1)]} × 100, in which A_1_ = AIA content in diet (% DM), A_2_ = AIA content in feces (% DM), F_1_ = nutrient content in diet (% DM) and F_2_ = nutrient content of feces (% DM).

### 2.5. Measurement of Enzyme Activities

Approximately 1 g of frozen jejunum and ileum tissue samples were weighed and homogenized with nine times the volume (wt/vol) of precooled physiological saline. The mixture was centrifuged at 2500× *g* for 10 min at 4 °C to collect the supernatant solution. The supernatant protein concentration was measured using commercial assay kits from Nanjing Jiancheng Biochemistry (Nanjing, China) according to the manufacturer’s instructions as the protein standard. The activities of trypsin, lipase and amylase in the supernatant solution were measured using commercial kits (Nanjing Jiancheng Bioengineering Institute, Nanjing, China) combined with a UV-VIS Spectrophotometer (UV1100, MAPADA, Shanghai, China) according to the manufacturer’s instructions.

### 2.6. Intestinal Morphology

The morphology measurements of the villus height and crypt depth were conducted [21]. briefly, 4-cm of each sample from middle sections of duodenum, jejunum, and ileum were washed with cold sterile saline and fixed with 4% paraformaldehyde solution. The mixed tissue samples were dehydrated with normal saline and then embedded in paraffin. The preserved samples were stained with hematoxylin and eosin. Ten well-orientated sections of height villi and their adjoint crypts in each sample were measured with Image Pro Plus software (Olympus Optical Company, Shenzhen, China) at 40× magnification.

### 2.7. Serum Hormone Parameters

Serum growth hormone (GH), Insulin, glucagon-like peptide-1 (GLP-1), leptin, ghrelin, cholecystokinin (CCK) and Peptide YY (PYY) levels were measured using commercially available porcine-specific ELISA kits (Jiangsu Jingmei Biotechnology Co., Ltd., Yancheng, China). All measurements were conducted in triplicate at minimum according to the manufacturer’s instructions. Each parameter was determined in triplicate simultaneously on the same plate. Additionally, the differences among parallels were required to be small (with a coefficient of variation less than 10%) to guarantee the reproducibility of repeated measurements.

### 2.8. DNA Extraction and Quantification of Intestinal Microflora

Microbial genomic DNA was isolated from the digesta samples (approximately 0.2 g) using the E.Z.N.A stool DNA kit (Omega Bio-Tek, Doraville, GA, USA) in accordance with the manufacturer’s protocols. Primers and probes (Table 2) for total bacteria *Escherichia coli*, *Lactobacillus*, *Bifidobacterium* and *Bacillus* were obtained from the previous work [22], which were commercially synthesized using Invitrogen (Shanghai, China). Quantitative real-time PCR was performed with a CFX96 Real-Time PCR Detection System (Bio-Rad Laboratories, Inc., Hercules, CA, USA). For determining the total bacteria, each measurement was run in a volume of 25 μL with 1 μL forward primer, 1 μL reverse primer, 12.5 μL SYBR Premix EX Taq (TaKaRa), 1 μL template DNA and 9.5 μL nuclease-free water. The thermal cycling conditions were an initial presaturation step at 95 °C for 15 min, 40 cycles of denaturation at 95 °C for 5 s, annealing at 59 °C for 25 s and extension at 72 °C for 60 s. For the quantification of *E. coli*, *Lactobacillus*, *Bifidobacterium* and *Bacillus*, real-time PCR was conducted in a volume of 20 μL with 1 μL probe enhancer solution, 0.3 μL probe, 1 μL forward and 1 μL reverse primer, 8 μL RealMasterMix (Tiangen, Beijing, China), 1 μL template DNA and 7.7 μL nuclease-free water. The reactions were subjected to 1 cycle at 95 °C for 15 min, followed by 49 cycles at 95 °C for 3 s, 58 °C for 25 s and 72 °C for 60 s. The cycle threshold (Ct) values and baseline settings were determined using automatic analysis settings and the copy numbers of the target group for each reaction were calculated from the standard curves, which were generated by constructing standard plasmids by a 10-fold serial dilution of plasmid DNA (1 × 10^1^ to 1 × 10^9^ copies/μL).

### 2.9. Microbial Metabolites Analysis

The concentrations of acetate, propionate and butyrate in the digesta of cecum and colon were determined via gas chromatography (CP-3800 GC, Varian, Inc., Walnut Creek, CA, USA), following the instructions described by Jiang et al. (Jiang et al., 2019).

### 2.10. Statistical Analysis

Descriptive statistics work was performed to evaluate whether data were normally distributed with statistical software SPSS 21.0 (IBM, USA). Then, a one-way ANOVA test was used to compare the difference of normally distributed data among groups, followed by Duncan’s multiple-range test. Results were presented as means and SEM. *P* < 0.05 was considered statistically significant, and 0.05 ≤ *P* ≤ 0.1 was considered a tendency.

## 3. Results

### 3.1. Quality of Fermented Feed

As shown in Table 3, due to changes in dry matter and organic matter, the content of calcium, phosphorus, lactic acid and acid-soluble protein increased after fermentation, while the content of crude fiber and pH in the feed decreased after fermentation.

### 3.2. Growth Performance

As shown in Table 4, during the 8–20 kg stage, compared with the CON group, the FLF group tended to increase the final BW of pigs (*P* < 0.1), significantly increased the ADG and ADFI of pigs (*P* < 0.05), and pigs fed the FLF diet significantly increased ADG and ADFI compared with the results from those fed the AB diet (*P* < 0.05). During the 20–50 kg stage, the FLF group significantly elevated ADG, ADFI and the final BW of pigs compared with that from the CON group (*P* < 0.05). In addition, the AB group significantly increased the ADG and ADFI of pigs (*P* < 0.05) compared with that from the CON group. There were no significant differences between the AB and FLF group (*P* > 0.1). During the 50–75 kg stage, compared with the CON group, the final BW and ADG of pigs fed the FLF diet were significantly increased (*P* < 0.05), and the ADG of pigs fed the AB diet was significantly elevated (*P* < 0.05). There were no significant differences between the AB and FLF group (*P* > 0.1). During the 75–100 kg stage, the FLF diet tended to increase the final BW compared with the CON diet (*P* < 0.1). There were no significant differences between the AB and FLF diets (*P* > 0.1). During the 100–125 kg stage, the final BW of pigs fed the FLF diet were greater than those fed the CON diet (*P* < 0.05), and the ADFI of pigs in the FLF and AB groups significantly increased compared with those of the CON group (*P* < 0.05). During the whole stage, the final BW and ADG of pigs fed the FLF diet were significantly increased compared with the those fed the CON diet (*P* < 0.05), and the ADFI of pigs fed the FLF diet tended to increase compared with that of those fed the CON diet (*P* > 0.05). There were no significant differences between the AB and FLF diets (*P* > 0.1).

### 3.3. Nutrient Digestibility

As shown in Table 5, during the 8–20 kg stage, the ATTD of DM, CP, EE, Ash, CF, GE, Ca and TP in the FLF group were significantly increased compared with those of the CON group (*P* < 0.05), and the ATTD of DM, CP, Ash, CF, GE and TP in the AB group were significantly elevated compared with those of the CON group (*P* < 0.05). The ATTD of DM, CP, Ash, CF, GE, Ca and TP in the FLF group were higher than those in the AB group (*P* < 0.05).

During the 20–50 kg stage, the ATTD of TP in the FLF group was significantly increased compared with that of the CON group (*P* < 0.05), and the ATTD of DM in the AB group was significantly elevated compared with that of the CON group (*P* < 0.05). The ATTD of TP in the FLF group was significantly increased compared with that of the AB group (*P* < 0.05).

During the 50–75 kg stage, the ATTD of EE and TP in the FLF group was significantly increased compared with that of the CON group (*P* < 0.05). The ATTD of TP in the FLF group was greater than that in the AB group (*P* < 0.05).

During the 75–100 kg stage, compared with the CON and AB groups, the ATTD of Ca in the FLF group tended to increase (*P* < 0.1), but there was no significant difference between the AB and FLF groups (*P* > 0.1).

During the 100–125 kg stage, the ATTD of EE, Ca and TP in the FLF group was significantly increased compared with that of the CON group (*P* < 0.05) and the ATTD of EE and TP in the FLF group was significantly increased compared with that of the AB group (*P* < 0.05); however, the ATTD of GE in the FLF group was significantly decreased compared with that of the CON and AB groups (*P* < 0.05).

### 3.4. Enzyme Activities

According to Table 6, there were no significant differences in the activities of the digestive enzymes of jejunum and ileum among the three groups (*P* > 0.1).

### 3.5. Serum Hormone Parameters

As shown in Figure 1, compared with that of the CON group, the level of serum leptin in the FLF group had a tendency to decrease (*P* < 0.1), the level of serum ghrelin was significantly increased (*P* < 0.05) and the level of serum PYY was significantly decreased (*P* < 0.05).

### 3.6. Intestinal Morphology

As shown in Table 7, there was no significant difference in the intestinal morphology of the duodenum, jejunum and ileum among the three groups (*P* > 0.1).

### 3.7. Intestinal Microbiota

As shown in Figure 2, the number of *Lactobacillus* of cecum and colon digesta in the FLF group was significantly increased compared with that of the CON group (*P* < 0.05), the number of *Escherichia coli* in the FLF group was significantly decreased compared with that of the CON group (*P* < 0.05), and the number of *Escherichia coli* of cecum digesta in the AB group was lower than that in the CON group (*P* < 0.05). Compared with that of the AB group, the number of *Lactobacillus* in cecum and colon digesta in the FLF group was significantly increased (*P* < 0.05), and the number of *Escherichia coli* was significantly decreased (*P* < 0.05).

### 3.8. Intestinal Microbial Metabolites

As shown in Figure 3, compared with the CON group, the level of acetic acid in the colonic digesta of the FLF group was significantly elevated (*P* < 0.05), and the total VFA in colonic digesta had a tendency to increase (*P* < 0.1). The acetic acid level in the colonic digesta of the FLF group was significantly increased compared with that of the AB group (*P* < 0.05).

## 4. Discussion

### 4.1. Quality of Fermented Feed

Lactic acid and acid-soluble protein are important indexes to measure the quality of fermentation, which can directly reflect the quality of fermented feed. This study showed that fermentation elevated the relative contents of lactic acid, acid-soluble protein, calcium and phosphorus in the feed and decreased the content of crude fiber and pH in the fermented feed, which were consistent with the results reported by previous studies [23,24].

### 4.2. Growth Performance

A large number of studies have shown that fermented feed or liquid feeding can improve the performance of pigs. Fermented feed can increase ADG [23,24] and ADFI [25] and reduce F/G of pigs [11,26]. In addition, liquid feeding also improved the growth performance of pigs, including in ADFI and ADG [4,22]. The results of this experiment are basically in line with those of previous studies. Possible reasons for the improvement in growth performance of fermented liquid feeding are as follows: (1) Fermented liquid feeding can increase the content of small peptides (such as acid-soluble protein [11]) and decrease the content of antinutritional factors, increasing the palatability and digestibility of feed [27]. (2) Fermented liquid feeding can effectively solve the problem of insufficient drinking water caused by competition among pigs and promote growth performance [28]. However, no effect on the ratio of feed to gain was observed in this experiment among the three groups. This is at variance with the findings in other studies, in which feed intake and growth rate were generally improved by liquid feeding at the expense of some reduction in F/G [29]. In the primary experiment, possible reasons for the lack of effect on the F/G of fermented liquid feeding might correlate with abundant feed waste caused by feed overflow in the feeder.

### 4.3. Nutrient Digestibility

Fermented feed and liquid feed can not only improve the growth performance, but also improve the ATTD of nutrients. Studies have shown that fermented feed improved the ATTD of nutrients of pigs [30,31,32]. However, only limited research on the effects of liquid feeding on the ATTD of pigs has been published. Previous study reported that liquid feeding improved the digestibility of DM and CP in pigs [33]. Alongside this, liquid fed growing-finishing pigs had a greater ATTD of P than that of dry fed pigs [34]. The results of this experiment were basically consistent with previous studies. During the nursery stage of weaned piglets, the ATTD of nutrients such as DM, GE and CP were significantly increased in the FLF group. The reason for this may be that fermented liquid feeding increases the level of small peptides in the feed [11] and increases the area of contact between digestive enzymes and nutrients [35]. In addition, the results of this experiment showed that the ATTD of minerals (especially P) was mainly increased in the FLF group during the growing and finishing stage of pigs, supporting the results reported previously [34,36].

### 4.4. Intestinal Enzyme Activities and Morphology

The primary study found that the FLF diet improved the growth performance and nutrient digestibility of pigs. The enzyme activities in the digestive tract were considered as important factors that would influence intestinal health and nutrient digestibility [9]. In order to further investigate whether the FLF diet had positive effects on intestinal digestive enzymes and intestinal morphology, we tested intestinal digestive enzymes and intestinal morphology indexes, and the results showed that there were no significant differences in intestinal enzymes and intestinal morphology among the three groups. Previous studies in weaned piglets showed that liquid feeding increased the activities of amylase and lipase in jejunum and improved intestinal morphology [22]. Meanwhile, fermented feed could increase the trypsin activity of weaned piglets [37]. The results of this study are different from those of previous studies. We speculate that finishing pigs have a more complete intestinal system than do weaned piglets [38]; therefore, fermented liquid feeding is not effective on the enzyme activities in the digestive tract of finishing pigs.

### 4.5. Serum Hormone Parameters

As the FLF diet did not affect the intestinal morphology and digestive enzymes of pigs, we speculated whether the FLF diet would promote the growth performance of pigs by regulating feeding hormones. Gastrointestinal hormone levels are closely related to food intake [39,40,41]. Leptin is a fat signal that regulates energy metabolism throughout the body, making it one of the best physiological markers of food intake [42,43]. Ghrelin is a neuropeptide that regulates feeding behavior in animals. Ghrelin has been implicated in both short-term and long-term appetite and body weight regulation [44,45,46]. PYY, an important intake regulatory hormone, can inhibit gastric acid secretion and gastrointestinal motility [47,48]. The results of this experiment showed that serum leptin and PYY levels were decreased and ghrelin level were increased in the FLF group. These hormonal changes may partially explain the increase in feed intake in the FLF group, which is consistent with the consequence of growth performance in this experiment. The results showed that the FLF diet could promote feed intake through the secretion of gastrointestinal hormones, improving the growth performance of animals.

### 4.6. Intestinal Microbiota and Microbial Metabolites

Beneficial bacteria in the intestines attach to the mucus layer of the intestines, inhibiting the adhesion and proliferation of harmful bacteria [49,50]. The disruption of intestinal microecological balance would affect the health and growth of animals [51,52]. A large number of studies have reported that fermented feed or liquid feeding improved the intestinal microorganism balance of pigs [53,54,55]. The results of this study showed that the FLF diet increased the number of *Lactobacillus* of cecal digesta, and decreased the number of *E. coli* of cecal digesta, which is consistent with the results of previous studies. The possible reasons for the positive effect on the intestinal microbiota of fermented liquid feeding might correlate with the low pH of a diet caused by large amount of lactate, preventing the development of *E. coli* [56]. Alongside this, our present study also indicated that the FLF diet increased concentration of acetic acid and total VFA in colonic digesta, which were consistent with the results of the population of *Escherichia coli*. Acetic acid content is negatively correlated with the number of *E. coli* bacterium in the intestine [57]. SCFAs can inhibit harmful bacteria through the increase in intercellular acidity in harmful bacteria, destroying the balance of osmotic pressure in harmful bacteria, thus playing an important role in the regulation of microflora [58]. In conclusion, the FLF diet can improve the intestinal microecology of pigs.

## 5. Conclusions

In summary, the results of the present study indicated that fermented liquid feeding improved the ADFI, ADG and ATTD of nutrients in pigs. Furthermore, fermented liquid feeding regulated the secretion of gastrointestinal hormones, improved intestinal chyme flora and increased volatile fatty acids in colon digesta. Therefore, fermented liquid feeding can improve the growth performance of pigs by regulating the secretion of gastrointestinal hormones and improving intestinal microecology.

## Figures and Tables

**Figure 1 animals-11-01452-f001:**
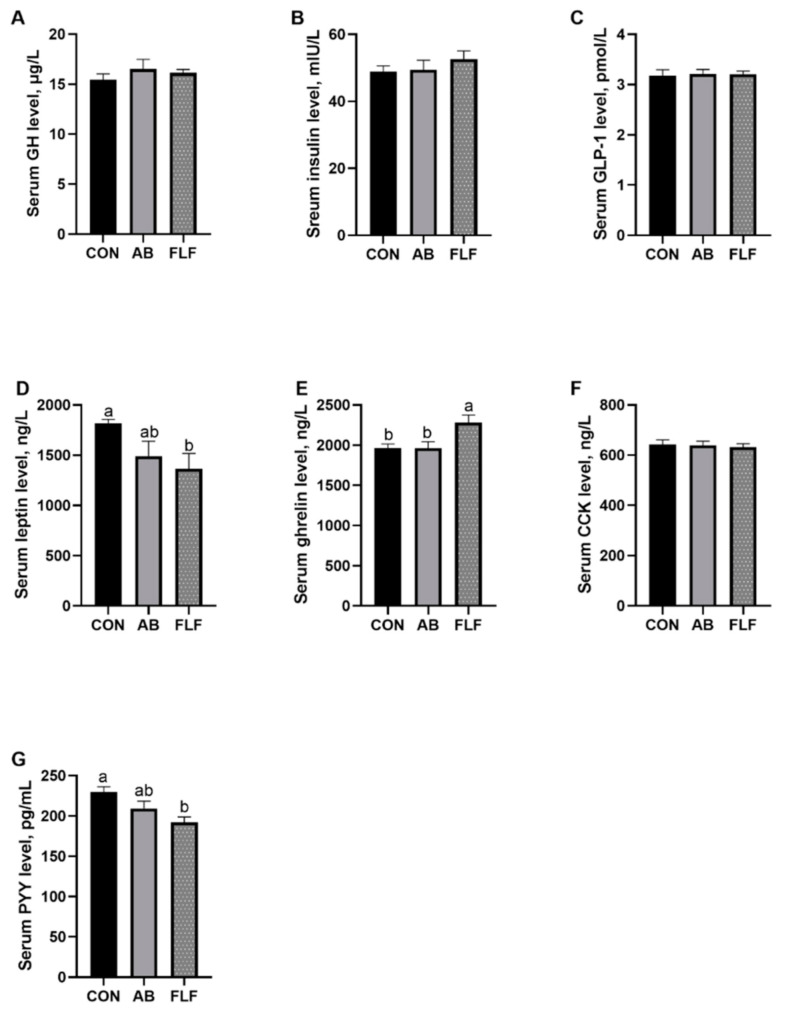
Effects of fermented liquid feeding on the serum hormone parameter of pigs. Values are means ± SEM, (*n* = 6). CON: control group; AB: antibiotics group; FLF: fermented liquid feeding group. Serum GH level (**A**); Serum insulin level (**B**); Serum GLP-1 level (**C**); Serum leptin level (**D**); Serum ghrelin level (**E**); Serum CCK level (**F**); Serum PYY level (**G**). GH: growth hormone; GLP-1: glucagon—like peptide 1; CCK: cholecystokinin; PYY: peptide YY. a, b Mean values with different letters on vertical bars indicate significant differences (*P* < 0.05).

**Figure 2 animals-11-01452-f002:**
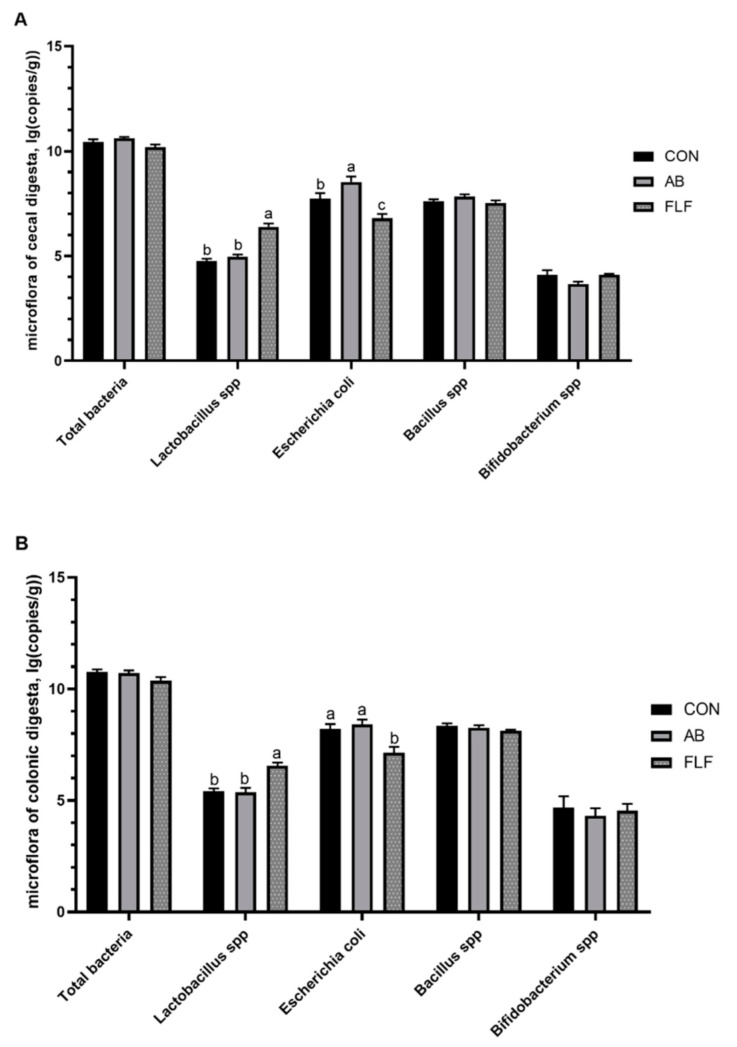
Effects of fermented liquid feeding on the contents of microflora in the cecal (**A**) and colonic (**B**) digesta of pigs (lg (copies/g)). Values are means ± SEM, (*n* = 6). CON: control group; AB: antibiotics group; FLF: fermented liquid feeding group. a, b, c mean values with different letters on vertical bars indicate significant differences (*P* < 0.05).

**Figure 3 animals-11-01452-f003:**
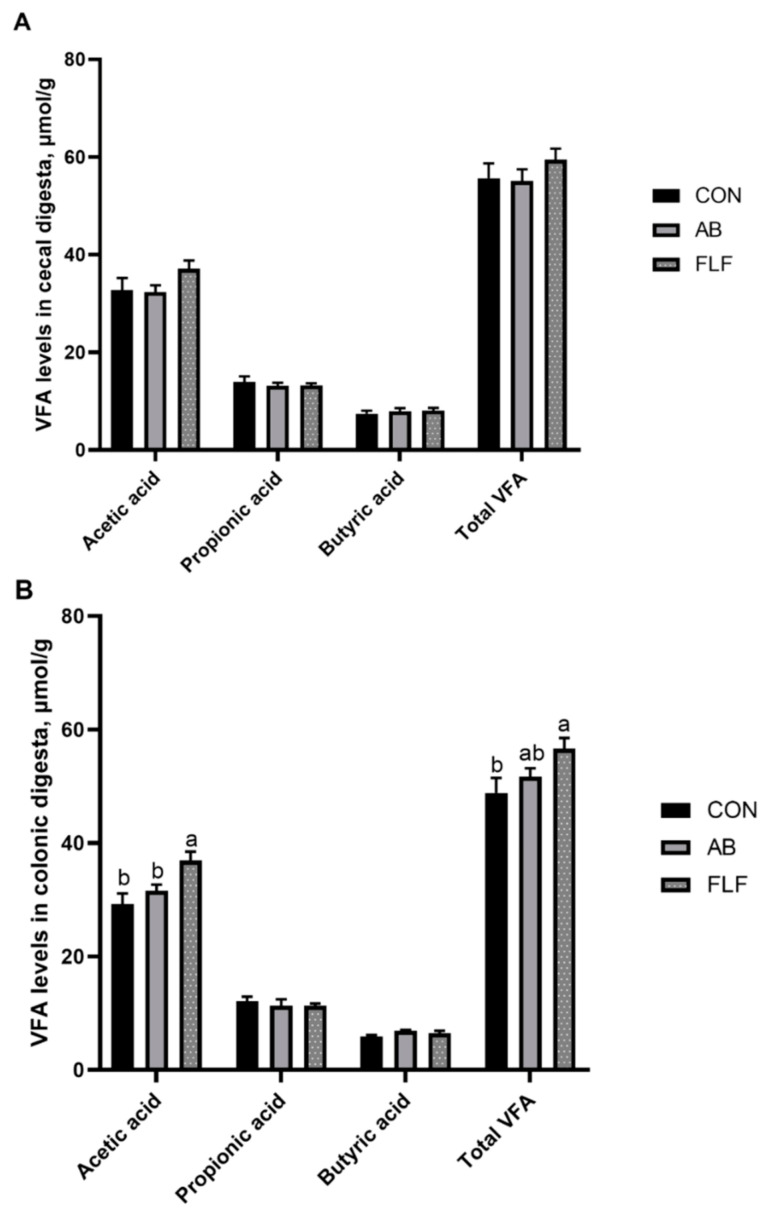
Effects of fermented liquid feeding on the levels of volatile fatty acids in the cecal (**A**) and colonic (**B**) digesta of pigs (μmol/g) Values are means ± SEM, (*n* = 6). CON: control group; AB: antibiotics group; FLF: fermented liquid feeding group. a, b mean values with different letters on vertical bars indicate significant differences (*P* < 0.05).

**Table 1 animals-11-01452-t001:** Composition and nutrient levels of basal diet (on an as-fed basis).

Ingredients	Phase (kg)				
	8–20	20–50	50–75	75–100	100–125
Extruded corn	30.00	10.00	0.00	0.00	0.00
Corn	30.23	57.11	68.14	77.21	73.36
Rice bran	0.00	2.00	4.00	0.00	4.00
Wheat bran	0.00	0.00	0.00	2.00	0.00
Phytase	0.00	0.00	0.00	0.01	0.00
Soy protein concentrate	4.00	0.00	0.00	0.00	0.00
Low protein whey powder	4.00	0.00	0.00	0.00	0.00
Extruded soybean	10.00	10.00	8.00	0.00	5.00
Soybean meal	10.00	13.10	15.00	18.00	13.00
Soybean oil	1.80	2.00	2.00	0.00	2.00
Fish meal	4.00	3.00	0.00	0.00	0.00
Sucrose	2.00	0.00	0.00	0.00	0.00
Glucose	1.00	0.00	0.00	0.00	0.00
Nacl	0.40	0.40	0.40	0.40	0.40
Chloride choline	0.18	0.15	0.15	0.15	0.15
Limestone	0.86	1.00	1.05	0.95	0.95
Dicalcium phosphate	0.52	0.40	0.52	0.59	0.45
Vitamin premix ^1^	0.05	0.04	0.04	0.04	0.04
Mineral premix ^2^	0.20	0.20	0.20	0.20	0.20
l-Lysine HCl	0.50	0.34	0.30	0.28	0.28
l-Threonine	0.13	0.10	0.08	0.08	0.08
DL-Methionine	0.10	0.12	0.10	0.07	0.07
Tryptophan	0.03	0.04	0.02	0.02	0.02
Total	100.00	100.00	100.00	100.00	100.00
Calculated nutrient compositions					
DE (Mcal/Kg)	3.54	3.51	3.48	3.33	3.47
CP, %	19.58	17.09	15.62	14.30	13.98
Ca, %	0.81	0.67	0.59	0.55	0.53
Total P, %	0.59	0.53	0.51	0.47	0.46
Available P, %	0.41	0.33	0.27	0.27	0.24
Lys, %	1.37	1.11	0.97	0.87	0.85
Met, %	0.48	0.39	0.33	0.28	0.28
Met + Cys, %	0.75	0.63	0.56	0.49	0.49
Thr, %	0.80	0.67	0.60	0.58	0.54
Trp, %	0.23	0.21	0.18	0.17	0.16

^1^ The premix provides following per kg diet: VA, 9000 IU; VD_3,_ 3000 IU; VE, 20 IU; VK_3,_ 3 mg; VB_1,_ 1.5 mg; VB_2,_ 4 mg; VB_6,_ 3.0 mg; VB_12,_ 0.2 mg; nicotinic acid, 30 mg; D-pantothenic acid, 15 mg; folic acid, 0.75 mg; biotin, 0.1 mg. ^2^ The premix provides following per kg diet: Fe, 100 mg as ferrous sulfate; Cu, 6 mg as zinc sulfate; Zn, 100 mg as zinc sulfate; Mn, 4 mg as manganese sulfate; I, 0.14 mg as potassium iodide; Se, 0.3 mg as sodium selenite.

**Table 2 animals-11-01452-t002:** Sequence of primers and probes used for the real-time PCR analysis of microbial populations.

Primer	Nucleotide Sequence (5′-3′)	A_T_, °C	Product Size, bp
Total bacteria	F: ACTCCTACGGGAGGCAGCAG	60	200
	R: ATTACCGCGGCTGCTGG		
*Lactobacillus*	F: ACTCCTACGGGAGGCAGCAG	60	126
	R: CAACAGTTACTCTGACACCCGTTCTTC		
	P: AAGAAGGGTTTCGGCTCGTAAAACTCTGTT		
*Escherichia coli*	F: CATGCCGCGTGTATGAAGAA	60	96
	R: CGGGTAACGTCAATGAGCAAA		
	P: AGGTATTAACTTTACTCCCTTCCTC		
*Bacillus*	F: GCAACGAGCGCAACCCTTGA	60	92
	R: TCATCCCCACCTTCCTCCGGT		
	P: CGGTTTGTCACCGGCAGTCACCT		
*Bifidobacterium*	F: CGCGTCCGGTGTGAAAG	60	121
	R: CTTCCCGATATCTACACATTCCA		
	P: ATTCCACCGTTACACCGGGAA		

F = forward primer; R = reverse primer; P = probe; A_T_ = annealing temperature.

**Table 3 animals-11-01452-t003:** Nutrient composition of basic feed and fermented feed.

Items	8–20 kg		20–50 kg		50–7 5 kg		75–100 kg		100–125 kg	
	CON	FLF	CON	FLF	CON	FLF	CON	FLF	CON	FLF
Dry matter, %	90.75	64.85	88.14	58.95	85.96	58.38	87.93	56.64	88.10	56.38
Crude protein, %	20.40	22.24	18.67	18.80	18.43	18.80	18.59	17.60	16.51	16.32
Crude fat, %	6.08	6.70	6.47	7.10	4.69	6.15	2.68	3.00	2.83	3.00
Crude fiber, %	1.78	1.50	2.18	1.50	2.80	2.10	1.76	1.53	2.57	2.36
Calcium, %	0.73	0.76	0.53	0.60	0.64	0.61	0.44	0.67	0.52	0.70
Total phosphorus, %	0.50	0.59	0.52	0.58	0.49	0.52	0.48	0.55	0.37	0.49
pH	6.90	4.02	6.92	4.01	6.89	4.04	6.91	4.01	6.93	4.03
Lactic acid, mmol/kg	35.62	67.22	61.81	84.72	84.03	105.14	75.97	102.64	23.06	63.19
Acid-soluble protein, %	1.24	3.15	1.60	2.98	1.24	3.05	1.17	2.46	1.17	1.78

CON: control group, FLF: fermented liquid feeding group.

**Table 4 animals-11-01452-t004:** Effects of fermented liquid feeding on the growth performance of pigs.

Items	CON	AB	FLF	SEM	*P*
8–20 kg					
Initial BW, kg	8.21	8.21	8.21	0.27	1
Final BW, kg	23.53	23.95	26.23	0.52	0.068
ADG, g	313 ^b^	321 ^b^	368 ^a^	7.83	0.002
ADFI, g	563 ^b^	570 ^b^	661 ^a^	14.38	0.002
F/G	1.8	1.78	1.8	0.02	0.798
20–50 kg					
Final BW, kg	46.04 ^b^	49.19 ^a,b^	52.34 ^a^	0.99	0.023
ADG, g	643 ^b^	721 ^a^	746 ^a^	15.99	0.013
ADFI, g	1504 ^b^	1679 ^a^	1714 ^a^	35.59	0.004
F/G	2.34	2.33	2.3	0.25	0.804
50–75 kg					
Final BW, kg	71.15 ^b^	76.62 ^a,b^	80.79 ^a^	1.42	0.01
ADG, g	718 ^b^	784 ^a^	813 ^a^	15.02	0.018
ADFI, g	2333	2621	2640	65.62	0.081
F/G	3.14	3.25	3.09	0.04	0.263
75–100 kg					
Final BW, kg	103.6	109.79	111.56	1.49	0.062
ADG, g	1159	11,895	1099	16.7	0.093
ADFI, g	3288	3379	3346	49.95	0.76
F/G	2.76	2.78	2.96	0.05	0.202
100–125 kg					
Final BW, kg	124.39 ^b^	132.14 ^a,b^	134.86 ^a^	1.78	0.032
ADG, g	1039	1118	1165	30.01	0.234
ADFI, g	3476 ^b^	4044 ^a^	3913 ^a^	99.14	0.007
F/G	3.37	3.63	3.35	0.09	0.466
8–125 kg					
Initial BW, kg Final BW, kg	8.21	8.21	8.21	0.27	1
ADG(g)	124.39 ^b^	132.14 ^a,b^	134.86 ^a^	1.78	0.032
ADFI, g	726 ^b^	775 ^a^	792 ^a^	10.09	0.012
F/G	1920	2097	2114	26.09	0.086
	2.64	2.71	2.67	0.26	0.624

*n* = 6. CON: control group, AB: antibiotics group, FLF: fermented liquid feeding group, ADG: average weight gain, ADFI: average daily feed intake, F/G: the ratio of feed to gain. ^a, b^ In the same row, values with different letter superscripts signal a significant difference.

**Table 5 animals-11-01452-t005:** Effects of fermented liquid feeding on the nutrition ATTD of pigs.

Items, %	CON	AB	FLF	SEM	*P*
8–20 kg					
Dry matter	79.08 ^c^	81.79 ^b^	88.26 ^a^	0.96	<0.05
Crude protein	70.27 ^c^	75.44 ^b^	84.40 ^a^	1.46	<0.05
Ether extract	68.25 ^b^	75.85 ^a^	74.35 ^a^	0.89	<0.05
Crude ash	34.45 ^c^	44.29 ^b^	60.57 ^a^	2.72	<0.05
Crude fiber	20.64 ^c^	30.13 ^b^	50.76 ^a^	3.29	<0.05
Gross energy	79.50 ^c^	82.20 ^b^	88.01 ^a^	0.89	<0.05
Calcium	42.45 ^b^	39.22 ^b^	64.96 ^a^	2.88	<0.05
Total phosphorus	22.33 ^c^	36.33 ^b^	75.54 ^a^	5.55	<0.05
20–50 kg					
Dry matter	86.80 ^b^	88.59 ^a^	87.66 ^a,b^	0.30	0.043
Crude protein	83.35	84.90	83.80	0.46	0.380
Ether extract	80.80	82.93	80.94	0.50	0.156
Crude ash	58.87	55.48	57.77	0.92	0.322
Crude fiber	58.86	58.64	58.02	0.55	0.826
Gross energy	88.29	89.12	88.14	0.20	0.088
Calcium	74.42	73.81	73.81	0.61	0.905
Total phosphorus	69.30 ^b^	66.39 ^b^	77.59 ^a^	1.43	<0.05
50–75 kg					
Dry matter	85.93	86.01	86.94	0.28	0.280
Crude protein	82.83	83.25	83.03	0.39	0.916
Ether extract	71.46 ^b^	73.99 ^a,b^	78.02 ^a^	1.01	0.017
Crude ash	58.87	58.23	58.58	0.67	0.936
Crude fiber	48.63	48.76	48.03	0.79	0.932
Gross energy	86.95	87.42	86.57	0.27	0.474
Calcium	61.74	61.26	62.82	0.70	0.676
Total phosphorus	58.67 ^b^	58.15 ^b^	72.38 ^a^	1.78	<0.05
75–100 kg					
Dry matter	89.98	89.72	90.72	0.11	0.134
Crude protein	88.83	88.80	88.12	0.18	0.182
Ether extract	64.75	65.95	64.98	0.43	0.506
Crude ash	60.96	61.17	62.55	0.45	0.314
Crude fiber	43.48	43.25	43.15	1.10	0.993
Gross energy	90.63	90.35	90.34	0.12	0.548
Calcium	64.91	64.50	68.89	0.91	0.086
Total phosphorus	73.03	71.08	72.39	0.58	0.248
100–125 kg					
Dry matter	90.14	90.42	89.80	0.17	0.368
Crude protein	86.39	87.02	86.08	0.30	0.423
Ether extract	58.60 ^b^	60.40 ^b^	78.03 ^a^	2.36	< 0.05
Crude ash	56.48	56.58	58.57	0.92	0.611
Crude fiber	59.79	61.42	61.65	0.87	0.660
Gross energy	91.22 ^a^	91.43 ^a^	90.02 ^b^	0.21	0.005
Calcium	64.58 ^b^	66.37 ^a,b^	70.69 ^a^	0.99	0.024
Total phosphorus	48.69 ^b^	55.58 ^b^	66.69 ^a^	2.24	<0.05

*n* = 6. CON: control group, AB: antibiotics group, FLF: fermented liquid feeding group. ^a, b^ In the same row, values with different letter superscripts mean significant difference.

**Table 6 animals-11-01452-t006:** Effects of fermented liquid feeding on digestive enzyme activities in the intestinal tissue of pigs.

Items	CON	AB	FLF	SEM	*P*
Jejunum					
Amylase, U/mgprot	3.46	3.02	3.6	0.26	0.673
Trypsin, U/mgprot	1271.78	1209.81	1206.27	48.52	0.839
Lipase, U/mgprot	16.96	16.24	16.29	0.39	0.725
Ileum					
Amylase, U/mgprot	2.86	2.95	2.81	0.14	0.933
Trypsin, U/mgprot	1100.68	1104.46	1164.68	75.95	0.935
Lipase, U/mgprot	15.68	16.17	16.99	0.9	0.843

*n* = 6. CON: control group, AB: antibiotics group, FLF: fermented liquid feeding group, mgprot = milligrams of protein.

**Table 7 animals-11-01452-t007:** Effects of fermented liquid feeding on intestinal morphology of pigs.

Items	CON	AB	FLF	SEM	*P*
Duodenum					
Villus height, µm	551.81	523.35	546.1	21.32	0.871
Crypt depth, µm	354.07	379.55	370.04	11.04	0.671
Villus: crypt	1.54	1.41	1.53	0.05	0.539
Jejunum					
Villus height, µm	541.07	451.98	517.95	18.23	0.129
Crypt depth, µm	264.59	226.42	278.26	11.76	0.181
Villus: crypt	1.95	2.13	1.94	0.07	0.521
Ileum					
Villus height, µm	515.72	520.74	508.75	20.15	0.977
Crypt depth, µm	314.44	325.87	326.82	11.24	0.905
Villus: crypt	1.64	1.6	1.66	0.04	0.613

*n* = 6. CON: control group, AB: antibiotics group, FLF: fermented liquid feeding group, Villus: crypt = villus height: crypt depth.

## Data Availability

The datasets used to support the findings of this study are available from the corresponding author upon request.

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
