# Peer review of "Fermented Diet Liquid Feeding Improves Growth Performance and Intestinal Function of Pigs"

_animals, 2021, doi:10.3390/ani11051452_

Round 1
Reviewer 1 Report
In the manuscript, author evaluated the effect of fermented liquid feeding on growth performance, apparent total track nutrient digestibility, and digestive enzymes and hormones in wean to finisher pigs.
It appears that FLF feeding improved growth performance when compare to pigs fed control diets. However, some detail information are required to compare antibiotics feeding with fermented liquid feeding.
Line 47: “In 2020, China's drug feed additives in pigs h have also been similarly banned in pig diets.”
Remove h and rephrase to “The same regulation was also issued by the Ministry of Agriculture and Rural Affairs of China In pigs in 2020”
Line 52 “often applied in global swine production[1], which”
Please provide the reference to proof this techniques being adopted in other countries.
Please rephase to “often applied in global swine production[1]. It…….”
Line 54 “high labor intensity and waste of liquid feeding mixed with hand is not conducive to its popularization, and”
This statement is not in agreement with Line 51.
Line 62 “They have been extensively 61 studied due to its benefits of increasing nutrient bioavailability and nutritional value [5].”
The reference cite is the meta analysis article with data mainly from Asia rather than EU?
Line 65” use of compound raw 65 materials “
Do you mean complete diet?
Line 81 “the strains (1mL/kg each, ≥108 CFU/mL)”
Please specify the Strains used.
Line 104: Please specify how long that pigs being fed with antibiotics.
Table 1. Please double check the percentage of inclusion rate of each ingredient. If the ingredient was not used, please remove from table.
Was the amino acids content determined for each diet? It seems fermented diet contained higher CP in nursery diet?
Line 112: How many times that pigs were fed in grower/finisher period?
Line 113: Was FLF prepared fresh for each meal?
Line 116 Was waste feed also accounted toward intake?
If it is so, was intake measurement based on dry matter basis?
Line 118 “All pigs were weighed at the beginning and the end of the experiment”
I believe it should be “All pigs were weighed at the beginning and the end of each phase”??
Line 123: How many pigs was housed per pen? If my calculation is correct, author collect fecal from all pigs from each pen? Is that right?
Line 219 “There were no significant differences between AB and FLF (P > 0.1).”
This doesn’t agree with results table ?
Line 228: again It doesn’t agree with table
Table 4 superscript used for P <0 .5 need to be differed from 0.05 < P <0.1
Also author should try to be consistent on superscript between traits
In addition, note that 75-100 kg is the only phase that wheat bran was replaced by rice bran, which appears have detrimental effect on ADG in FLF. Does this mean the effect of FLF on growth performance is ingredient dependent?
Line 246: ATTD TP from pigs fed FLF was higher than pigs fed AB?
Line 252: P value for ATTD TP is 0.248? Perhaps it is calcium rather than TP?
Line 253: It doesn’t agree with table. The superscript indicates that there is not significant different between pigs fed AB and FLF on ATTD of Ca?
Table 5 Please double check P value ATTD of GE and TP in 100-125 kg was showed approach significant but no superscript was assigned.
Line 300: “In the primary experiment, possible reasons for no effect on F/G of fermented liquid feeding 300 might correlate with abundant feed waste caused by feed overflow in the feeder.”
Did feed waste being accounted toward feed intake?
Line 305 Please discuss the differences between liquid feeding, fermented feed and fermented liquid feeding.
Author Response
Dear Editor and reviewers,
Thank you very much for your precious comments and advice to improve our manuscript.
Response to Reviewer #1
Comments:
Point1. It appears that FLF feeding improved growth performance when compare to pigs fed control diets. However, some detail information are required to compare antibiotics feeding with fermented liquid feeding.
Response 1: Thank you for your suggestion. We revised the paper according to your precious comment. We have shown the comparison between the fermented liquid feeding group and the antibiotics group in writings.
Point 2. Line 47: “In 2020, China's drug feed additives in pigs h have also been similarly banned in pig diets.
Response 2: Thank you for your suggestion. We revised the paper according to your precious comment.
Point 3. Line 52 “often applied in global swine production[1], which ”Please provide the reference to proof this techniques being adopted in other countries. Please rephase to “often applied in global swine production[1]. It…….”
Response 3: Thank you for your suggestion. We revised the paper according to your precious comment.
Point 4. Line 54 “high labor intensity and waste of liquid feeding mixed with hand is not conducive to its popularization, and”
Response 4: Thank you for your suggestion. We revised the paper according to your precious comment.
Point 5. Line 62 “They have been extensively 61 studied due to its benefits of increasing nutrient bioavailability and nutritional value [5].”
Response 5: Thank you for your suggestion. Article data source description “We systematically searched in PubMed and Web of Science for studies (published between January 1st, 2000 and December 20th, 2018) comparing the effects of FF supplementation with basal diet on pig growth performance.”
Point 6. Line 65” use of compound raw 65 materials “
Response 6: Thank you for your suggestion. We revised the paper according to your precious comment. What we want to express is the complete diet.
Point 7. Line 81 “the strains (1mL/kg each, ≥108 CFU/mL)”
Response 7: Thank you for your suggestion. We revised the paper according to your precious comment.
Point 8. Line 104: Please specify how long that pigs being fed with antibiotics.
Response 8: Thank you for your suggestion. We revised the paper according to your precious comment. The feeding time of all treatment groups was 160 days.
Point 9. Table 1. Please double check the percentage of inclusion rate of each ingredient. If the ingredient was not used, please remove from table. Was the amino acids content determined for each diet? It seems fermented diet contained higher CP in nursery diet?
Response 9: Thank you for your suggestion. We revised the paper according to your precious comment. The basal diet was formulated to meet or exceed the nutrient requirements recommended by the NRC (2012). The composition of the fermented diet at each stage is exactly the same as that of the basic group.
Point 10. Was the amino acids content determined for each diet? It seems fermented diet contained higher CP in nursery diet?
Response 10: Thank you for your suggestion. The basal diet was formulated to meet or exceed the nutrient requirements recommended by the NRC (2012). The composition of the fermented diet at nursery stage is exactly the same as that of the basic group.
Point 11. Line 112: How many times that pigs were fed in grower/finisher period?
Response 11: Thank you for your suggestion. All pigs were fed diets 4 times per day at 0800, 1200, 1600, 2000, for a 160-d period.
Point 12. Line 113: Was FLF prepared fresh for each meal?
Response 12: Thank you for your suggestion. FLF was prepared fresh for each meal.
Point 13. Line 116 Was waste feed also accounted toward intake? If it is so, was intake measurement based on dry matter basis?
Response 13: Thank you for your suggestion. Waste feed was also accounted toward intake and was intake measurement based on dry matter basis.
Point 14. Line 118 “All pigs were weighed at the beginning and the end of the experiment”I believe it should be “All pigs were weighed at the beginning and the end of each phase”??
Response 14: Thank you for your suggestion. We revised the paper according to your precious comment. All pigs were weighed at the beginning and the end of each phase.
Point 15. Line 123: How many pigs was housed per pen? If my calculation is correct, author collect fecal from all pigs from each pen? Is that right?
Response 15: Thank you for your suggestion. Every pen housed 16 pigs. Fecal was collected from all pigs from each pen.
Point 16. Line 219 “There were no significant differences between AB and FLF (P > 0.1).”This doesn’t agree with results table ?Line 228: again It doesn’t agree with table.Table 4 superscript used for P <0 .5 need to be differed from 0.05 < P <0.1.Also author should try to be consistent on superscript between traits.
Response 16: Thank you for your suggestion. We revised the paper according to your precious comment. The results are consistent with the data in the table.
Point 17. In addition, note that 75-100 kg is the only phase that wheat bran was replaced by rice bran, which appears have detrimental effect on ADG in FLF. Does this mean the effect of FLF on growth performance is ingredient dependent?
Response 17: Thank you for your suggestion. Maybe you are right, We have not found relevant research support, which may be used as an entry point for further research.
Point 18. Line 246: ATTD TP from pigs fed FLF was higher than pigs fed AB?
Response 18: Thank you for your suggestion. We revised the paper according to your precious comment.
Point 19. Line 252: P value for ATTD TP is 0.248? Perhaps it is calcium rather than TP?
Response 19: Thank you for your suggestion. We revised the paper according to your precious comment. That should be Ca, not TP.
Point 20. Line 253: It doesn’t agree with table. The superscript indicates that there is not significant different between pigs fed AB and FLF on ATTD of Ca?
Response 20: Thank you for your suggestion. We revised the paper according to your precious comment.
Point 21. Table 5 Please double check P value ATTD of GE and TP in 100-125 kg was showed approach significant but no superscript was assigned.
Response 21: Thank you for your suggestion. We revised the paper according to your precious comment.
Point 22. Line 300: “In the primary experiment, possible reasons for no effect on F/G of fermented liquid feeding 300 might correlate with abundant feed waste caused by feed overflow in the feeder.”Did feed waste being accounted toward feed intake?
Response 22: Thank you for your suggestion. We consider the factor of wasted feed when calculating feed intake and f/G.
Point 23. Line 305 Please discuss the differences between liquid feeding, fermented feed and fermented liquid feeding.
Response 23: Thank you for your suggestion. We revised the paper according to your precious comment. Liquid feeding is a feeding method in which feed and water are fully mixed according to a certain ratio before feeding; The feed made from roughage through microbial fermentation is fermented feed; Fermented liquid feeding means that the feed is fermented and then mixed with a certain proportion of water before feeding.
Thank you again for your valuable and very helpful comments. We would be glad to respond to any further questions and comments that you may have.
Yours sincerely,
Ping Zheng

Reviewer 2 Report
General opinion. The topic is not a new one. There are some points which have to be improved. English language has to be corrected.
There is no information whether the Animal Ethics Committee approved the experimental protocol.
The advantage of this manuscript is a range covering different periods of pig growth (fattening) and indicators assessing the influence of experimental factors described here.
Introduction is rather general and does not convince the reader to follow the text of manuscript. Authors used KITASAMYCIN as a feed antibiotic. It is not popular very much except from China and some introduction of it should be made.
It is not true that are no studies devoted to the use of fermented diets in the finishing pig's feeding.
Results. It seems rather impossible that "fermentation increased the contents of Calcium, phosphorus...". This is rather the effect of dry matter content.
Discussion. In many cases it is comparison with other published data. More explanation of achieved results would be welcome.
Conclusions. The authors should remember that general conclusion about role of feed fermentation is drawn on a base of single experiment with only one type of mixture. However all periods of pigs fattening were analysed.
Tables. In any table the accuracy of digits should be changed. ADG, ADFI (g) should be given in integers as well as all values greater than 100 (or according to the Editor).
1. Lines where the material was not included to the diet (0) should be removed. There is also a question: what for the 2% of wheat bran was used in the feed for pig 75-100 kg?
Isn't it possible to present "metabolisable energy" instead of DE?
3. Gross energy does not differ in general feeds for pigs, so this value is out of importance.
Author Response
Dear Editor and reviewers,
Thank you very much for your precious comments and advice to improve our manuscript.
Response to Reviewer #2
Comments:
Point 1. There is no information whether the Animal Ethics Committee approved the experimental protocol.
Response 1: Thank you for your suggestion. We revised the paper according to your precious comment. Experimental procedure and animal care were accomplished in accordance with the guide for the care and use of laboratory animals provided by the Institutional Animal Care Advisory Committee for Sichuan Agricultural University.
Point 2. is rather general and does not convince the reader to follow the text of manuscript. Authors used KITASAMYCIN as a feed antibiotic. It is not popular very much except from China and some introduction of it should be made.
Response 2: Thank you for your suggestion. We revised the paper according to your precious comment. Kitasamycin is a multi-component macrolide antibiotic produced by Streptomyces. It has an inhibitory effect on gram-positive bacteria and gram-positive bacteria, especially for most penicillin and red mold Vegetarian Staphylococcus aureus is effective.
Point 3. It is not true that are no studies devoted to the use of fermented diets in the finishing pig's feeding.
Response 3: Thank you for your suggestion. We revised the paper according to your precious comment. We means “there has no research on fermented liquid feeding in finishing pigs has been reported, not the use of fermented diets in the finishing pig's feeding.
Point 4. It seems rather impossible that "fermentation increased the contents of Calcium, phosphorus...". This is rather the effect of dry matter content.
Response 4: Thank you for your suggestion. We revised the paper according to your precious comment. The minerals in the full-price compound feed should be conserved before and after fermentation, but the relative content of minerals after fermentation is significantly increased. This is because the fermentation consumes the organic matter of the full-price compound feed and the dry matter is reduced, resulting in the proportion of Ca, P in fermentation products has increased.
Point 5. In many cases it is comparison with other published data. More explanation of achieved results would be welcome.
Response 5: Thank you for your suggestion. We revised the paper according to your precious comment.
Point 6. The authors should remember that general conclusion about role of feed fermentation is drawn on a base of single experiment with only one type of mixture. However all periods of pigs fattening were analysed.
Response 6: Thank you for your suggestion. We revised the paper according to your precious comment.
Point 7. In any table the accuracy of digits should be changed. ADG, ADFI (g) should be given in integers as well as all values greater than 100 (or according to the Editor).
Response 7: Thank you for your suggestion. We revised the paper according to your precious comment.
Point 8. Lines where the material was not included to the diet (0) should be removed. There is also a question: what for the 2% of wheat bran was used in the feed for pig 75-100 kg?
Response 8: Thank you for your suggestion. We revised the paper according to your precious comment. 2% of wheat bran was used in the feed for pig during 75-100 kg. During the trial period, Novel Coronavirus Outbreak caused transportation difficulties. However, we ensured that the composition of the rations processed at each stage was consistent.
Point 9. Isn't it possible to present "metabolisable energy" instead of DE?
Response 9: Thank you for your suggestion. We use DE in the preparation of feed, but after calculation, the formula also meets the needs of ME.
Point 10. Gross energy does not differ in general feeds for pigs, so this value is out of importance.
Response 10: Thank you for your suggestion. We revised the paper according to your precious comment.
Thank you again for your valuable and very helpful comments. We would be glad to respond to any further questions and comments that you may have.
Yours sincerely,
Ping Zheng

Round 2
Reviewer 1 Report
Line 101 please add “All pigs remained on the same dietary treatments throughout the trial”
Line 103 The diet table still content ingredient that was not used in the trial. Please remove any ingredient that was not used for the trial.
Line 103 Was the amino acids content determined for each diet?
Point 12. Line 113: Was FLF prepared fresh for each meal?
Response 12: Thank you for your suggestion. FLF was prepared fresh for each meal.
Please add this statement in materials and methods
Line 216 “And FLF significantly elevated ADG, ADFI of pigs compared with AB (P < 0.05).”
Change to “and pigs fed FLF significantly increased ADG and ADFI compared with AB (P < 0.05).”
During 100 - 223
125 kg stage, the final BW of pigs in FLF were greater than those of CON (P < 0.05), and ADFI of pigs in FLF and AB tends to increase compared with CON (P < 0.05).
According to table it was statistic significant on ADFI of 100-125 kg period? Please verify
Also please add superscript to ADFI at 100-125 kg period
Please make sure the superscript is consistent across manuscript. Either assign the lowest mean with a or highest mean with a.
Line 233 During the whole stage, the final BW and ADG of pigs in FLF were significantly increased compared with the CON (P < 0.05), and the ADFI of pigs in FLF tends to increase compared with the CON (P >0.05).
Please define the tendency. 0.05 < P <0.1?
Please add superscript to those traits showed the tendency toward significant different
Author Response
Dear Editor and reviewers,
Thank you very much for your precious comments and advice to improve our manuscript.
Response to Reviewer #1
Comments:
Point1. Line 101 please add “All pigs remained on the same dietary treatments throughout the trial”
Response 1: Thank you for your suggestion. We revised the paper according to your precious comment. (line 102).
Point 2. Line 103 The diet table still content ingredient that was not used in the trial. Please remove any ingredient that was not used for the trial.
Response 2: Thank you for your suggestion. We revised the paper according to your precious comment. (Table 1).
Point 3. Was FLF prepared fresh for each meal? Please add this statement in materials and methods
Response 3: Thank you for your suggestion. FLF was prepared fresh for each day. We revised the paper according to your precious comment. (line 116).
Point 4. Line 216 “And FLF significantly elevated ADG, ADFI of pigs compared with AB (P < 0.05).”Change to “and pigs fed FLF significantly increased ADG and ADFI compared with AB (P < 0.05).”
Response 4: Thank you for your suggestion. We revised the paper according to your precious comment. (line 224-225).
Point 5. During 100 - 125 kg stage, the final BW of pigs in FLF were greater than those of CON (P < 0.05), and ADFI of pigs in FLF and AB tends to increase compared with CON (P < 0.05). According to table it was statistic significant on ADFI of 100-125 kg period? Please verify. Also please add superscript to ADFI at 100-125 kg period. Please make sure the superscript is consistent across manuscript. Either assign the lowest mean with a or highest mean with a
Response 5: Thank you for your suggestion. We revised the paper according to your precious comment. We check to make sure that the superscripts are consistent across the manuscripts (line 234-236).
Point 6. Line 233 During the whole stage, the final BW and ADG of pigs in FLF were significantly increased compared with the CON (P < 0.05), and the ADFI of pigs in FLF tends to increase compared with the CON (P >0.05).
Please define the tendency. 0.05 < P <0.1?
Response 6: Thank you for your suggestion. We revised the paper according to your precious comment. We check to make sure that the superscripts are consistent across the manuscripts, and we define the tendency is 0.05 ≤ P <0.1(line 236-240).
Thank you again for your valuable and very helpful comments. We would be glad to respond to any further questions and comments that you may have.
Yours sincerely,
Ping Zheng

Reviewer 2 Report
The explanations given to the reviewer are not in the text (e.g. kitasamycin) - at least I couldn't find it.
It can not be stated that "fermentation increased the Calcium and Phosphorus content" - I have written about it before. It can be said that: "because of changes in dry matter and organic matter, the content of calcium and phosphorus increased after fermentation.
I wonder what a reason is to give (e.g. Table 4) the data with such accuracy - ADFI - 1504.41; 2322.54 .... The values following the point are not significant for evaluation of the results. They only reduce the readability of the table. In my opinion it should be integers in such situation.
Author Response
Dear Editor and reviewers,
Thank you very much for your precious comments and advice to improve our manuscript.
Response to Reviewer #2
Comments:
Point 1. The explanations given to the reviewer are not in the text (e.g. kitasamycin) - at least I couldn't find it.
Response 1: Thank you for your suggestion. We revised the paper according to your precious comment. We have add the explanations for kitasamycin in the text. (line 107-110).
Point 2. It can not be stated that "fermentation increased the Calcium and Phosphorus content" - I have written about it before. It can be said that: "because of changes in dry matter and organic matter, the content of calcium , phosphorus increased after fermentation.
Response 2: Thank you for your suggestion. We revised the paper according to your precious comment. (line 221-224).
Point 3. I wonder what a reason is to give (e.g. Table 4) the data with such accuracy - ADFI - 1504.41; 2322.54 .... The values following the point are not significant for evaluation of the results. They only reduce the readability of the table. In my opinion it should be integers in such situation.
Response 3: Thank you for your suggestion. We revised the paper according to your precious comment. (table 4).
Thank you again for your valuable and very helpful comments. We would be glad to respond to any further questions and comments that you may have.
Yours sincerely,
Ping Zheng
